# In silico voltage-sensitive dye imaging reveals the emergent dynamics of cortical populations

Taylor H. Newton [1,2✉], Michael W. Reimann [1], Marwan Abdellah[1], Grigori Chevtchenko[1],
Eilif B. Muller[1,3,4,5] & Henry Markram[1,6]

Voltage-sensitive dye imaging (VSDI) is a powerful technique for interrogating membrane potential dynamics in assemblies of cortical neurons, but with effective resolution limits that confound interpretation. To address this limitation, we developed an in silico model of VSDI in a biologically faithful digital reconstruction of rodent neocortical microcircuitry. Using this model, we extend previous experimental observations regarding the cellular origins of VSDI, finding that the signal is driven primarily by neurons in layers 2/3 and 5, and that VSDI measurements do not capture individual spikes. Furthermore, we test the capacity of VSD image sequences to discriminate between afferent thalamic inputs at various spatial locations to estimate a lower bound on the functional resolution of VSDI. Our approach underscores the power of a bottom-up computational approach for relating scales of cortical processing.

[1] Blue Brain Project, École Polytechnique Fédérale de Lausanne (EPFL), Geneva, Switzerland. [2] IT'IS Foundation for Research on Information Technologies in Society, Zurich, Switzerland. [3] Department of Neurosciences, Faculty of Medicine, University of Montreal, Montreal, QC, Canada. [4] CHU Sainte-Justine Research Center, Montreal, QC, Canada. [5] Quebec Artificial Intelligence Institute (Mila), Montreal, QC, Canada. [6] Laboratory of Neural Microcircuitry, Brain Mind Institute, EPFL, Lausanne, Switzerland. ✉email: taylor.h.newton@gmail.com

Electrical signaling in cortex is thought to be divided into "inputs" in the form of subthreshold synaptic potentials, and "outputs" in the form of action potentials (APs) or spikes[1]. Therefore, a complete understanding of cortical function requires not only a means of recording spikes in neural ensembles but also a technique for resolving subthreshold membrane potentials ($V_m$) in these populations. Voltage-sensitive dye imaging (VSDI) is a mesoscale imaging technique capable of capturing subthreshold activity across the entire rodent neocortical surface (on the order of several cm$^2$) with good spatiotemporal resolution (on the order of milliseconds in time, and <50 μm in space).[1–4]

In previous decades, significant progress was made in mapping the functional architecture of the brain using intrinsic optical imaging, a modality based on activity-dependent changes in the intrinsic absorptive and fluorescent properties of brain tissue. Studies combining intrinsic optical imaging with cytochrome oxidase staining revealed the interdependent organization of ocular dominance columns, cytochrome oxidase blobs (color preference), and orientation-selective "pinwheel" structures in the visual system[5–9]. However, intrinsic optical imaging is limited by a slow time constant (on the order of seconds[10,11]), rendering it ill-suited for capturing temporal changes in ongoing activity. To this end, VSDI has added a dynamic component to the understanding of neural assemblies. For example, VSDI-based studies have clarified the role of feedforward thalamic inputs versus intracortical recurrent activity in shaping orientation selectivity[12], and shown how orientation-selective responses spread over the cortex as a function of stimulus shape and size[13]. VSDI has also been widely applied to the study of somatosensory computations in barrel cortex, where the somatotopic organization and spatiotemporal scale of activity are well suited to the technique. Such studies have produced important findings regarding the regulation of response dynamics by ongoing spontaneous activity[14], cortical state[15], behavior[16–18], and stimulus strength[19]. Broadly put, VSDI has enabled the field to move beyond the static picture provided by staining and intrinsic optical imaging, adding a dynamic dimension to the understanding of mesoscale cortical organization.

However, VSDI suffers from the limitation that the superposed activity of neurites belonging to many cells is reflected in each image pixel. Uneven dye penetration, and blurring due to the absorption and scattering of photons in tissue further complicate the interpretation of VSDI signals[1,2,20]. Indeed, a historical concern has been identifying which attributes of neural anatomy and physiology (e.g., layer, cell type, dendrites vs. axons, pre- vs. postsynaptic activity) are the primary drivers of VSDI measurements[16,19,21,22]. There exist previous efforts to construct detailed, bottom-up models of VSDI with the aim of answering these questions[23]. We build on the precedent set by such studies, and elaborate further on the relationship between signaling at the cellular level and dynamics at the level of neural populations.

Here, we present the results of a detailed computational model of VSDI, implemented in a digital reconstruction of the neocortical microcircuitry (NMC) of the hindlimb somatosensory cortex of a juvenile rat[24]. The NMC comprises a network of ~31,000 neurons with detailed cellular anatomy and physiology and data-driven synaptic physiology, and algorithmically constrained connectivity in a 0.29 ± 0.01 mm$^3$ column of tissue (Fig. 1a, b). To simulate VSDI signals, we performed simulations of the NMC to obtain $V_m$ recordings of neural compartments under various experimental conditions (Fig. 1c) (see also Supplementary Fig. 2 for an evaluation of the evoked VSDI response in several different instantiations of the NMC based on the anatomy of individual rodents). Next, we corrected the compartment voltages to account for the effects of dye penetration and light transport in cortical tissue (Fig. 1d), and collected this data into voxels (Fig. 1d, e).

Using offline Monte Carlo simulations of photon-tissue interactions and a ray transfer model of microscope optics, we calculated a depth-dependent point spread function (PSF), with which we convolved horizontal slices of voxelized data to account for optical blurring (Fig. 1e, f) (see also "Methods"). This procedure generated a time-ordered collection of VSD images (Fig. 1g), which we repeated for various permutations of microcircuit geometry (Fig. 1h) to probe population dynamics in the NMC model. To this end, we began by testing our model's capacity to recapitulate established results in the VSDI literature regarding the dynamics of evoked cortical activity spread. Next, we investigated several common but difficult to test assumptions concerning the cellular origins of VSDI signals: the insensitivity of VSDI measurements to individual spikes, and the respective contributions of distinct subpopulations of neurons. Finally, we leveraged our model's biophysical detail to predict the fundamental limits of VSDI's ability to resolve spatially distinct cortical inputs.

## Results

**Evoked VSDI response dynamics.** Propagating waves of activity are thought to support the representation and integration of information in cortex[16,17,19,25–28], and can be observed with VSDI. To quantify the similarity between the evoked response dynamics of our model and those reported in literature, we conducted a series of whisker flick-like trials (see "Methods") and examined the spread of activity. Our stimulation protocol consisted of a single pulse of activity in 60 contiguous thalamocortical (TC) fibers emanating from a virtual ventral posteromedial nucleus (VPM) projecting to the geometric center of a concentric arrangement of seven NMCs (i.e., the "mosaic", Fig. 1h).

We observed a radially expanding pattern of activation centered around the location of stimulus delivery, which expanded to fill the entire NMC surface over the course of several tens of milliseconds, reaching peak fluorescence at ~57 ms poststimulus (Fig. 2a, b). The peak was immediately followed by a period of declining activity characterized by increasing hyperpolarization, which undershot baseline fluorescence, reaching a minimum at 170 ms and gradually recovering to within 10% of baseline after ~510 ms. To quantify the temporal persistence of the signal, we calculated the half-width duration (decay time to 50% of peak with respect to baseline, 88 ms). The time to peak and half-width duration are comparable to the findings of[16], who report values of ~45 ms, and 86 ± 69 ms, respectively (Fig. 2b). We also considered the relationship between the instantaneous firing rate (computed using 3 ms bins) and the VSDI signal in a 100 ms poststimulus window (Fig. 2c, d). Our simulations indicate that peak AP firing occurred ~7 ms *prior* to peak VSD fluorescence, contrary to the intuition that increased mean $V_m$ precipitates population spiking.

Notably, a differing VSD activity time-course was observed at the stimulus delivery location relative to the NMC periphery (Fig. 2g). At peripheral points along the *x*- and *z*-axes (+540 μm and +460 μm, respectively), the rising and falling phases of the fluorescence response were almost identical to the spatial mean. In contrast, signal recorded at the stimulus location exhibited an initial transient within the first 12 ms of stimulus onset, and then gradually rose to peak fluorescence. This response pattern (initially confined, expanding thereafter) was also visible in the spatial profile of activation over time as an initially sharp peak, which gradually rose and then flattened into a plateau (Fig. 2f).

In order to characterize the propagation velocity of the evoked activity wavefront, we fit each frame in the image sequence to a two-dimensional Gaussian profile and measured the change in the full width at half maximum over time (Fig. 2e) (see

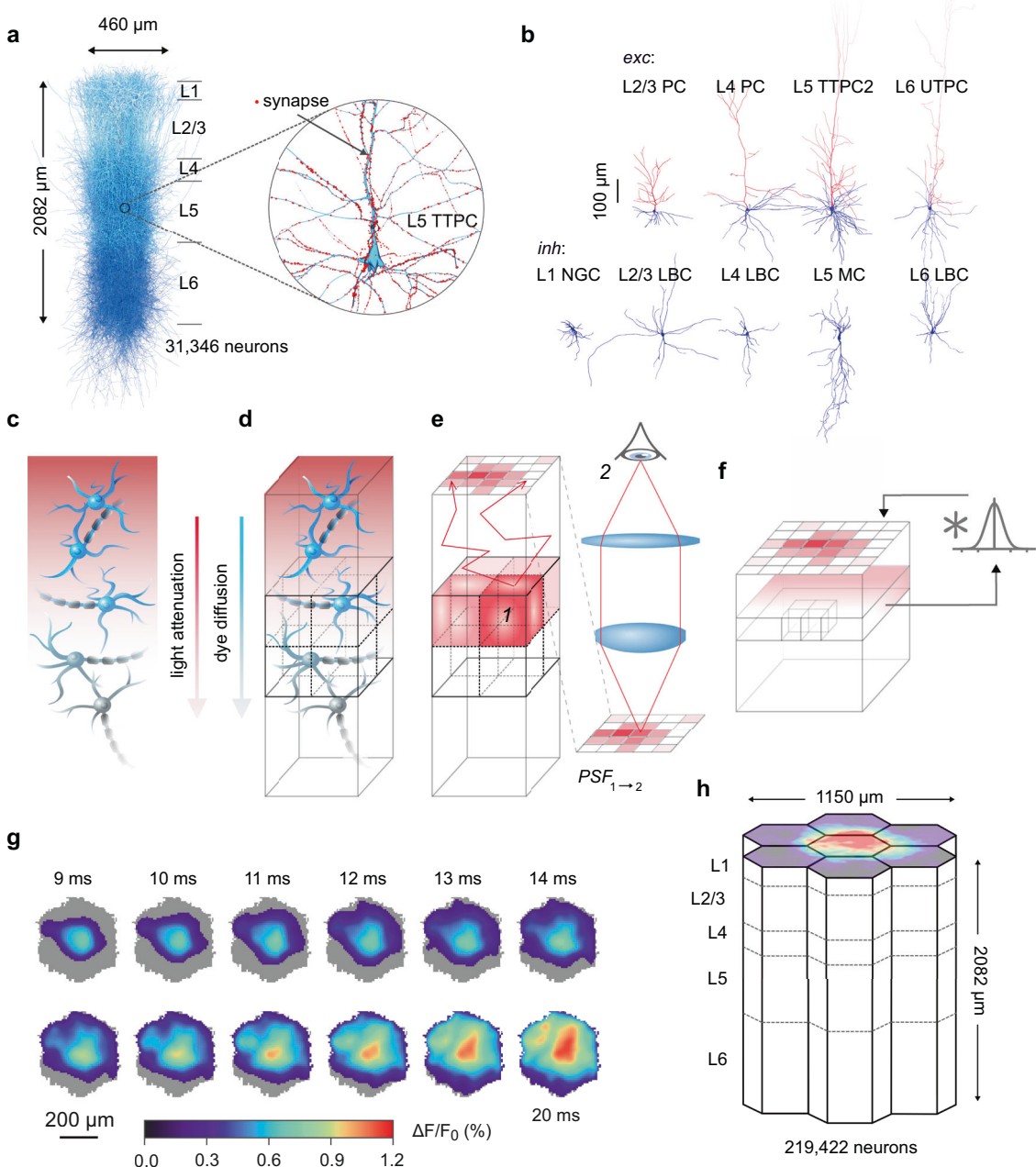

**Fig. 1 Cortical microcircuit overview and in silico VSDI workflow. a** Digital microcircuit comprising 31,346 morphologically detailed neurons connected in a columnar unit. Inset: expanded view of a L5 PC, with synapses highlighted in red. **b** Exemplar excitatory (top) and inhibitory (bottom) cell types. Blue: axons. Red: dendrites. **c–f** Schematic illustrating the in silico VSDI workflow. **c** Neuron surface areas are scaled by prefactors accounting for dye diffusion and light transport in cortical tissue. **d** Microcircuit volume is divided into voxels to facilitate calculations. **e** Photons emitted from each voxel are scattered and absorbed throughout the tissue volume via Monte Carlo simulations. Photons reaching the cortical surface are propagated through a tandem-lens optical setup using ray transfer matrix analysis. Steps **e** and **f** are performed once for a given circuit and optical setup to determine a depth-dependent point-spread function (point-spread from 1 to 2 in panel **e**, i.e., from voxel to camera). **f** Raw signals at each depth are convolved with their respective point-spread function, and accumulated in a pixel array at the surface. **g** Example VSDI image stack for 11 ms of spontaneous activity. Images were thresholded at 10% of peak response. **h** Microcircuits were aggregated into a larger volume made of a central microcircuit column surrounding by six additional columns contacting each of the central column's hexagonal sides (i.e., the "mosaic"). This arrangement mitigates boundary effects within the central column and facilitates the analysis of signal spread dynamics.

Supplementary Algorithm 1 for details). We found that activity wavefronts underwent two sequential bursts of expansion prior to peak VSD fluorescence, reaching a peak velocity of ~20 μm/ms. Subsequently, the wavefront entered a period of contraction (−10 μm/ms) near the fluorescence peak, before gradually returning to baseline (fluctuations near zero).

In vivo VSDI experiments have documented wavefront propagation speeds within an order of magnitude of those reported above. For example[14], use a Gaussian fit of the cross-sectional profile of VSD images to estimate that whisker deflection-evoked waves in urethane or halothane anesthetized rodent barrel cortex propagate along barrel rows at a speed of

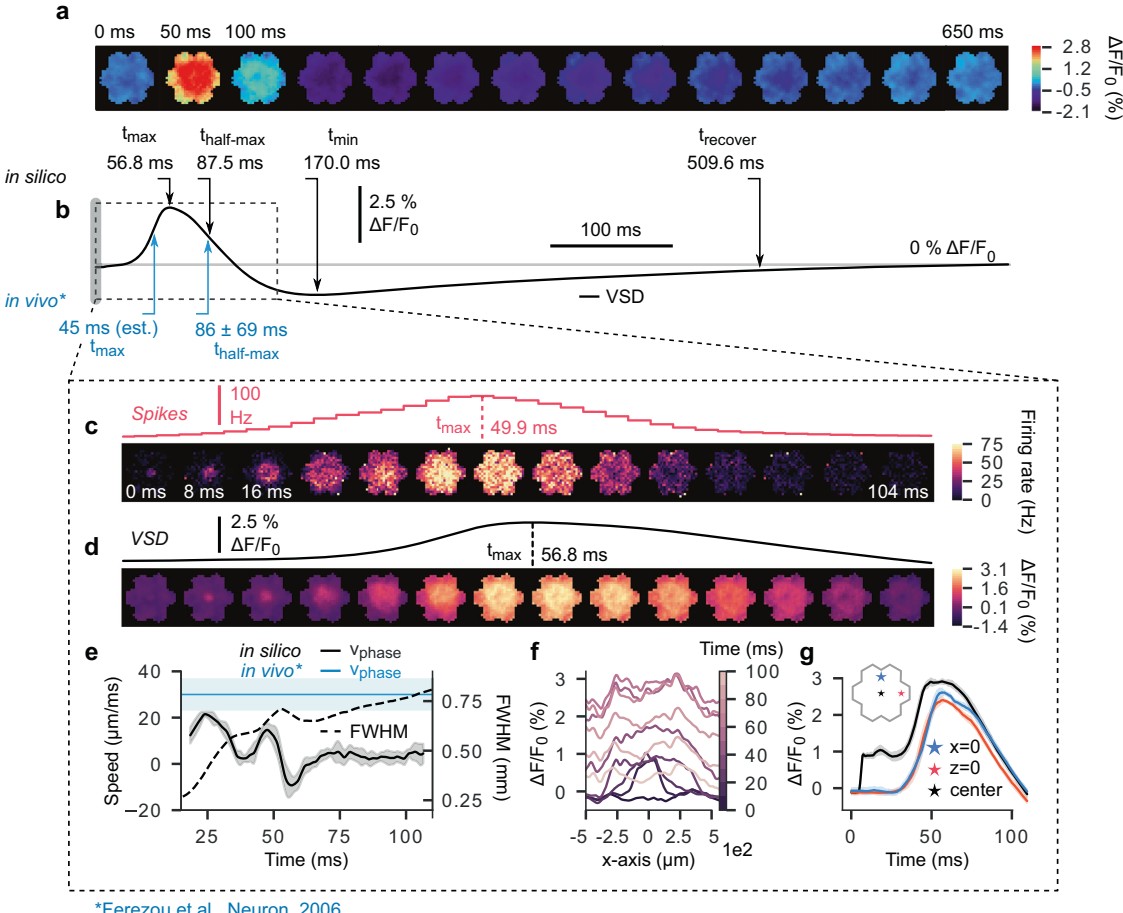

**Fig. 2 Propagation of stimulus-evoked cortical activity. a** 650 ms of stimulus-evoked VSD measurements in the "mosaic". The stimulus consists of a single, coincident pulse of activity in 60 contiguous thalamic projection fibers innervating the center of the interior microcircuit (delivered at $t = 0$ ms). **b** Time course of spatially averaged VSD signal in **a**. Upper arrows (black) indicate points of interest along the curve. From left to right: peak latency, half-maximum duration (time to decay to 50% of signal peak), time of signal minimum, recovery time (earliest time after minimum for amplitude to stably decay to within 10% of baseline). Bottom arrows (blue) indicate in vivo values for peak latency and half-maximum duration reported in literature. **c** Top: PSTH of spiking activity for time window (dashed box) indicated in **b** (3 ms bins). Bottom: pixel-wise PSTH (mean firing rate of all cells under each pixel) over same time window (8 ms bins). **d** Top: expanded view of time window (dashed box) indicated in **b** detailing ascending and descending phases of stimulus-evoked spatially averaged VSD signal. Bottom: same as above, but for spatially extended VSD signal, where each frame was computed by averaging activity in 8 ms intervals. **e** Activity wavefront propagation velocity (solid line, left axis), and wavefront size (full-width at half-maximum, dashed line, right axis). Blue horizontal line and shaded region show in vivo measurements of stimulus-evoked wavefront propagation velocity reported in literature. Solid blue line: mean value; shaded blue region: measurement range; shaded gray region: standard deviation ($n = 10$ independent trials). **f** Spatiotemporal evolution of VSD signal along the x-axis ($z = 0$, i.e., a horizontal line across the surface of the "mosaic"). Lightening hue represents the passage of time. **g** Time course of VSD activation at specific locations on the cortical surface. Blue star: circuit periphery along the y-axis. Red star: circuit periphery along the x-axis. Black star: circuit center. Colored error bands: standard deviation (n=10 independent trials).

$\sim$60 μm/ms, and barrel arcs at $\sim$33 μm/ms. See Supplementary Table 1 for a summary of cortical wavefront propagation velocities reported in the literature. Also, see Supplementary Fig. 3 for an analysis of laminar VSDI activity spread in a sagittal slice (x–y plane) of the NMC.

**Excitatory neurons in layers 2/3 and 5 dominate VSDI measurements.** VSDI signals are linearly proportional to the product of local $V_m$ and membrane surface area[1]. Moreover, signals originating in neurites located in deeper layers are significantly more attenuated than those emanating from superficial layers due to uneven dye penetration and light-tissue interactions (Fig. 3b). It follows that the morphology, location, and orientation of a given cell affect the magnitude of its contribution to the optical response. To better understand these influences, we analyzed the fractional contributions of cortical layers and cell types to the

overall VSDI signal. In agreement with previously reported results[14,17,22,29], we found that >90% of the raw fluorescence originated within 500 μm of the pial surface (Fig. 3b). Furthermore, we saw that neurites belonging to L2/3 and L5 neurons monopolized the "effective surface area", which we define as the quantity that results from multiplying the original surface area of each neurite by a depth-dependent scaling factor accounting for dye penetration and light transport; L2/3 and L5 contributed 44.9% and 43.7% of the total, respectively. As predicted by the distribution of effective surface area, L2/3 and L5 were also the primary drivers of the VSDI signal (47.8% and 37.6%, respectively, $n = 10$ trials) during spontaneous activity (Fig. 3c, d). Cross-correlation revealed mutual positive correlations between the contributions of each layer and the VSDI total (Fig. 3e). However, for evoked activity, L5 generated upwards of 67% of the signal whereas L2/3 neurites constituted 19% (Fig. 3f, g). Importantly, L5 underwent strong depolarization in the

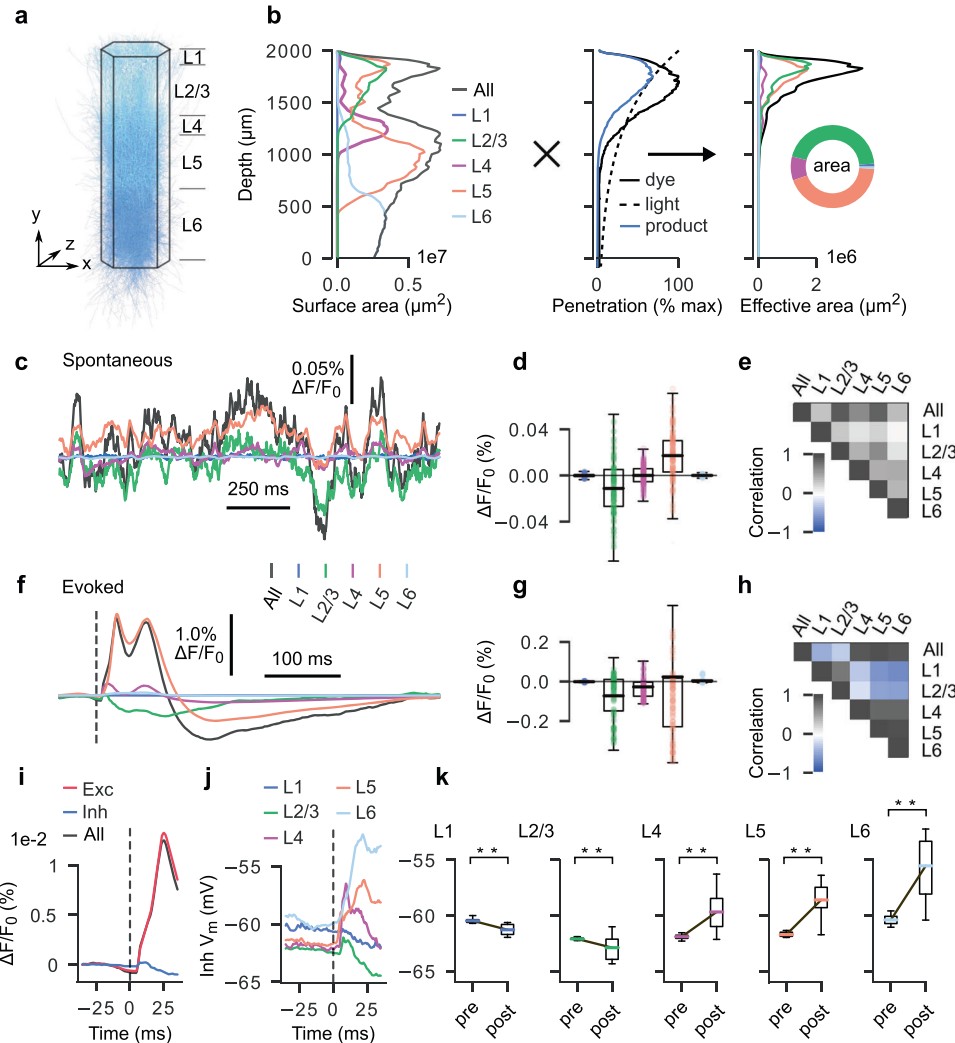

**Fig. 3 Fractional contributions to VSDI measurements by layer and cell-type. a–b** Surface area contributions for each layer by depth. **a** Microcircuit, for reference respecting relative layer positions and axis orientation. **b** Left: raw (unscaled) neurite surface area profiles by depth for each layer (20 μm bins). Middle: depth-dependent scaling prefactors accounting for dye diffusion (solid black line) and light penetration (dashed black line). Solid blue line indicates product. Right: effective surface area profiles by depth for each layer (raw surface area **b** scaled by product of light attenuation and dye diffusion prefactors (blue line, middle panel), 20 μm bins). **c** Spatially averaged VSD signal (black) with fractional contribution of each layer (colored) for 1.5 s of spontaneous activity. **d** Boxplot of fractional layer-wise contribution data in **c**, illustrating overall spread and polarity of each layer's contributions. Whiskers indicate standard deviation. $n = 30{,}010$ time samples acquired over $n = 10$ independent trials (3001 per trial; 2000 ms to 3500 ms in 0.5 ms steps). Box extends from lower to upper quartile values of the data, with a line at the median. Top and bottom whiskers indicate extent of data up to last datum below $Q3 + 1.5 \times IQR$, and last datum above $Q1 - 1.5 \times IQR$, respectively, where Q1 is the 1st quartile, Q3 is the 3rd quartile, and IQR is the interquartile range $(Q3 - Q1)$. **e** Correlation matrix for all traces in **c**. **f** Spatially-averaged VSD signal (black) with fractional contribution of each layer (colored) for 500 ms of evoked activity. Plot begins at $-50$ ms, stimulus delivered at 0 ms (dashed line). **g** Same as in **d** but for evoked activity. $n = 30010$ time samples acquired over n=10 independent trials (3001 per trial; 2000 ms to 3500 ms in 0.5 ms steps). Box and whisker definitions same as in **e**. **h** Same as in **e** but for evoked activity. **i** Fractional contributions of excitatory and inhibitory populations to overall VSD signal, shown over a 50 ms second window spanning 25 ms pre- and 25 ms poststimulus. **j** Mean membrane potentials computed for inhibitory cell populations in each layer, plotted over the same time window as in **i**. **k** Boxplots depicting the difference between pre- and poststimulus membrane potentials for inhibitory cell populations in each layer. Boxplots in each panel were calculated using the 25 ms pre- and poststimulus periods referred to in **i** and **j**, **\*\***: $p < 1e-10$, paired sample t test (two-sided). $n = 350$ time samples acquired over $n = 1$ trial (data for each time point calculated by averaging over inhibitory cell voltages per layer). Box and whisker definitions same as in **e**.

poststimulus window while L2/3 tended to hyperpolarize, indicating differential, layer-specific roles during stimulus-driven responses. Analysis of the correlations between layer contributions supports this conclusion, showing anticorrelated activity between superficial and deep layers (Fig. 3h). Finally, we note that these experiments (Fig. 3) were conducted in an isolated NMC column. In contrast, our analysis of cortical activation spread (Fig. 2) was computed in a mosaic concatenation of seven NMCs to allow sufficient surface area for signal spread. Concatenating

several columns slightly alters the lateral distributions of neurites, and accounts for the differences in peak activity profiles (monophasic vs. biphasic) and peak amplitudes (~2.5% vs. ~1%) apparent in Figs. 2b and 3f.

Morphological reconstructions indicate that the dendritic arbors of inhibitory neurons tend to be spatially confined relative to their excitatory counterparts (Fig. 1b; red: dendrites, blue: axons), which may influence their respective contributions to the VSDI signal. Thus, we also decomposed VSD fluorescence into

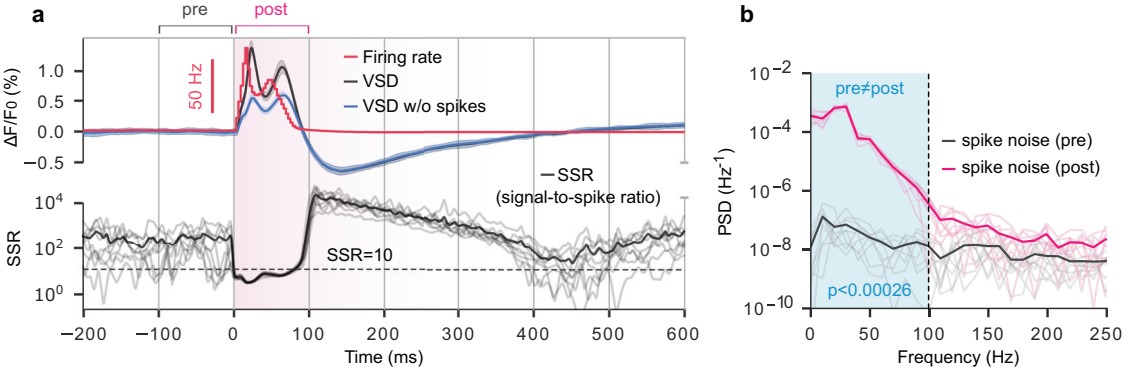

**Fig. 4 Detectability of spiking activity in the VSDI signal. a** Top: firing rate (red), VSD (black), and VSD computed excluding spikes (membrane potentials thresholded at −55 mV, blue) for 200 ms of spontaneous activity followed by 600 ms of evoked activity. Bottom: signal-to-spike ratio. Noise here refers to "spike noise", or contamination of putative subthreshold measurements by APs. Pale black lines are individual trials; dark line indicates mean. Dashed black line: signal-to-spike ratio = 10. **b** Power spectral density of spike noise computed for 100 ms prestimulus window (black), and 100 ms poststimulus window (magenta). Blue shaded box indicates frequencies for which pre- and poststimulus noise are significantly different. Dashed black line indicates frequency at which prestimulus noise and poststimulus noise are no longer meaningfully different (paired *t* test (two-sided), significance threshold = 0.01 adjusted to 2.6e-4 for multiple comparisons using Holm-Bonferroni correction).

excitatory and inhibitory components. For evoked trials, the excitatory component of the signal (>90%) underwent large deflections in the poststimulus window, far outweighing inhibitory contributions (<10%) (Fig. 3i). Indeed, the inhibitory fraction remained small throughout both pre- and poststimulus periods. One might expect the inhibitory VSDI fraction to increase in proportion to the excitatory fraction, as it is known that excitatory activity in healthy neocortex quickly recruits a mitigating inhibitory response, preventing runaway excitation[30–33]. We therefore analyzed the timecourse of mean membrane potential changes in the inhibitory populations of each layer, revealing that those in superficial layers were significantly hyperpolarized following stimulation, while those in deep layers were significantly depolarized (Fig. 3j, k). Thus, the spatially confined dendrites of inhibitory neurons located in deeper layers contributed significantly less to the overall signal as their arbors do not extend to a height reachable by dye and excitation light. As a consequence, only the contributions of hyperpolarized superficial inhibitory neurons are visible to the VSDI signal.

**Disentangling the impacts of sub- and suprathreshold neural activity on VSDI.** It is thought that APs are too brief and too asynchronous to contribute substantially to the VSDI signal, despite causing large fluctuations in $V_m$[1,14–16,19,34]. This conclusion is based on simultaneous VSDI and single-cell patch-clamp recordings, of which spike-triggered averaging exposes the absence of individual AP waveforms from the VSDI signal[16]. However, such experiments leave open the possibility that large *volleys* of spikes occurring within a narrow time window could still measurably influence the signal. We tested this conjecture by comparing spike-related VSDI contributions in pre- and post stimulus time windows lasting several hundred milliseconds each (Fig. 4). To isolate the effects of spikes on the optical response, we ran our VSDI pipeline on spike-filtered compartment voltage data and compared with unfiltered data (Fig. 4a). Assuming the null hypothesis that VSDI primarily reflects subthreshold activity, we considered any difference between the raw and spike-filtered signals as "noise" due to spikes. This allowed us to calculate a signal-to-spike ratio (SSR), defined in analogy to the signal-to-noise ratio, as the squared quotient of the root mean square amplitudes of the unfiltered signal and spiking component. That

is,

$$A = \sqrt{\frac{1}{\Delta t} \int_{t_0}^{t_1} (\text{VSD(t)})^2 \, dt} \qquad (1.1)$$

$$\text{SSR} \triangleq \left( \frac{A_{\text{raw}}}{A_{\text{raw-filt}}} \right)^2 \qquad (1.2)$$

which we represented as a continuous variable by binning into 40 ms intervals with overlapping windows. Conservatively, we estimate that typical VSDI experiments have an SNR of ~10[22,35–39]. Therefore, when SSR is <10 (i.e., less than the empirical SNR of typical experiments), the component of the VSDI signal due to spikes is larger than contamination due to other noise sources, and in principle could be detected. Although SSR did dip slightly below our estimated detectability threshold during the poststimulus window, this is unlikely to be meaningful in most laboratory settings. However, in cases where exceptionally high SNR is achieved, information regarding the spiking component of the VSDI signal may become relevant. Therefore, we sought to understand how the frequency content of spike noise is affected by stimulation (Fig. 4b).

An analysis of the power spectral density of spike noise immediately pre- and poststimulus showed that the frequency content differs significantly only below ~100 Hz, with lower frequencies exhibiting greater divergence. Measurements sensitive enough to detect a spiking contribution to the VSDI signal, therefore, would only contain spike-related information below this frequency cutoff and would be dominated by low-frequency components. It is important to acknowledge that, as reported previously, contributions by individual spikes are not detectable in mesoscale recordings; rather, it is the aggregate influence of population spiking that adds a small DC offset (and low-frequency oscillations) to the VSDI signal, as described above.

Assuming a high SNR scenario, we also undertook an analysis of the relative contributions of forward- and backward-propagating APs to the spike-related VSDI signal component (Supplementary Fig. 3). We found that nearly all of the spike-related VSDI signal is due to backward-propagating APs in dendritic arbors.

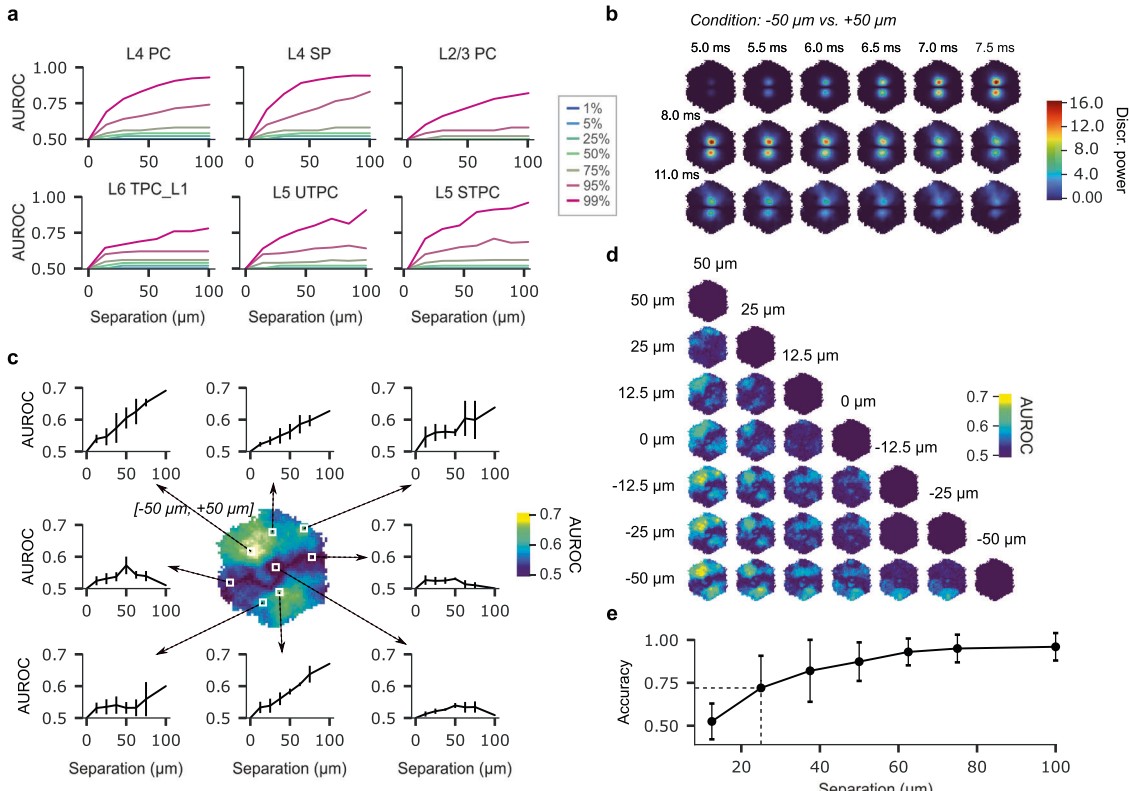

**Fig. 5 VSDI pixel-wise two-point discrimination of VPM input locations.** Various measures of circuit activity were compared for their ability to discriminate between VPM input stimulus locations. **a** Spike trains of individual neurons were used to compute AUROC scores (representing capacity to differentiate between stimulus locations) as a function of stimulus separation. Colors in legend indicate percentile of discrimination capacity within each subpopulation. **b** Discrimination power at discrete time points for stimuli positioned at −50 μm and +50 μm along a vertical axis bisecting the VSD image plane. Pixels are the negative log $p$ value of the null hypothesis that the VSDI data associated with each stimulus location come from the same distribution. **c** Center: summary map of discrimination performance (AUROC score) considering the entire time course of each VSDI pixel for stimuli at −50 μm and +50 μm. Surrounding plots: AUROC score as a function of increasing stimulus separation at several pixel locations indicated by white boxes (mean values, $n = (7, 8, 10, 8, 6, 4, 4, 2)$ independent scores per separation (0, 12.5, 25, 37,5, 50, 62.5, 75, 100 μm, respectively) reflecting the number of possible stimulus location pairings at that separation). Error bars indicate standard deviation for each separation. **d** Matrix of AUROC score maps for each combination of stimulus locations. **e** XGBoost-based classification accuracy as a function of stimulus separation considering all pixels at all time points (mean values, $n = (20, 25, 20, 15, 10, 10, 5)$ independent scores per separation (12.5, 25, 37,5, 50, 62.5, 75, 100 μm, respectively), reflecting the number of possible (unique) stimulus location pairings at that separation evaluated with fivefold cross-validation). Dashed lines indicate threshold (25 μm, ~75% accuracy) below which classification performance drops considerably. Whiskers indicate standard deviation.

**VSDI two-point discrimination.** The signal in each VSDI pixel reflects the summated activity of many dendritic processes belonging to many different neurons whose somas are distributed both horizontally, away from a given pixel, and vertically, throughout all six cortical layers. We have shown previously how such contributions break down by layer and neuron class. Now we consider how the multiplexed nature of VSDI signals could lead to loss of information.

First, as a baseline, we examined the capacity of the spike trains of individual neurons to discriminate between different stimuli; spiking activity constitutes the "output" communicated by the circuit to other brain regions and is therefore useful as a proxy for information content. To this end, we defined an experimental paradigm wherein we subjected the NMC to simulated inputs from the VPM nucleus of the thalamus at various spatial locations. As our model contains diverse sources of biological noise (Nolte et al.[40]), responses can vary significantly between repetitions of a given experiment, making the distinction between pairs of stimuli a non-trivial problem. Nevertheless, we found that spike trains of individual neurons can discriminate spatially separated stimuli with high accuracy, although performance depended strongly on the individual. Only the top 5% of neurons

achieved a level of accuracy at or exceeding 90%, and nearly half of those sampled never rose above chance levels (Fig. 5a).

As noted previously, individual spiking activity is effectively invisible to VSDI measurements, but the postsynaptic potentials (PSPs) resulting from volleys of stimulus-induced spikes may leave a detectable signature, albeit strongly mixed with PSPs elicited by neighboring neurons not directly contacted by thalamic projections. To investigate how evoked spiking activity impacts the VSDI response, we quantified the degree to which stimulation affects the value of a given pixel at a given point in time. Specifically, for pairs of spatially separated but otherwise identical stimuli, we calculated the negative logarithm of the null hypothesis that the distributions of pixel values have identical means ($t$ test, $n = 25$ repetitions of both stimuli). Figure 5b depicts a map of this measure for the first ~13 ms following stimulation, comparing delivery at −50 μm and +50 μm. We found that two regions with strongly differentiated VSDI responses emerge at 5 ms post stimulus, and gradually fade by 13 ms, coinciding with the initial appearance and subsequent dispersal of activity in the underlying VSDI datasets. Thus, pixels directly above TC projection fibers exhibit the strongest difference in the milliseconds immediately following stimulation.

Furthermore, we observed a horizontal band of very small difference along the axis of symmetry dividing the two stimulus locations, implying that pixels there undergo a similar activity trajectory for either stimulus condition. Having characterized the spatiotemporal evolution of differences in the VSDI signal for different stimuli, we proceeded by investigating how well individual pixels are able to discriminate between stimulus locations when considering the entire history of pixel values. Accordingly, we calculated the area under the receiver operating characteristic curve (AUROC) of the classification problem (see "Methods", and Fig. 5c). We found that overall discrimination performance is reduced as compared to spike train-based classification, reaching a maximum AUROC score of ~0.7 for high-performing pixels. As before, we found that some pixels provided no useful information for classification (AUROC score near 0.5). In addition, we found that certain pixels achieved a peak score at sub-maximal stimulus separations, although the trend showed an increase in pixel-wise AUROC score for increasing stimulus separation (Fig. 5c). Overall, discrimination performance for single pixels suffered in comparison to stimulus classification using only the spike trains of individual neurons. We therefore performed a final classification, this time using information from all pixels for a 700 ms time window including the stimulus presentation. This allowed us to predict a lower limit below which VSDI images lack the fundamental information content necessary to reliably discriminate between stimulus locations, i.e., an effective input resolution. For the classification, we selected "XGBoost", an implementation of the gradient boosting decision tree algorithm noted for its speed and accuracy on structured datasets[41] (see "Methods"). Using this method, we reached classification accuracies approaching 100% for large separations, indicating that the values of pixels contained non-redundant information. We further found that discrimination performance declined for smaller separations, dropping sharply at approximately 25 µm (Fig. 5e). We interpret this value as a lower bound on the effective input resolution of VSD imaging, accounting for both the aggregate nature of the signal and optical distortions introduced by photon scattering and absorption in the tissue. In most laboratory settings, additional sources of noise will also contaminate the signal, further compromising discrimination between input stimulus locations. Thus, future advances in imaging technology are unlikely to improve VSDI's fundamental capacity to resolve loci of cortical activity below this limit.

## Discussion

We constructed a bottom-up, biophysically detailed model of VSDI in a digital reconstruction of rodent somatosensory cortex to relate cellular anatomy and physiology to mesoscale signals. As a first step, we considered VSDI measurements of evoked responses in our model and found that they were qualitatively and quantitatively similar to analogous in vivo experiments. Next, we used our model to deconstruct the VSDI signal into layer and cell type contributions, revealing context-dependent, strongly differentiated roles for layers 2/3 and 5. We also examined the influence of spiking activity, and found that while individual spikes are not reflected in VSDI data, large volleys of semi-synchronous spikes could affect measurements. Finally, a two-point discrimination test of "functional" resolution showed that cortical VSDI is unlikely to be able to resolve point-like inputs separated by <25 µm. Several details pertaining to the above results merit further discussion.

**Lateral spread dynamics of VSDI signals**. As stated previously (see "Results"), the VSD fluorescence response to TC stimulation

underwent non-uniform lateral expansion, wherein the signal quickly saturated near the location of stimulus delivery before gradually extending across the entire microcircuit (Fig. 2f, g). This observation is consistent with in vivo optical responses to evoked activity, in which VSDI signals saturate in a locally confined region near the stimulation site within the first 10–20 ms before expanding outwards[15,19,21,42,43].

Fehérvári et al.[42] report similar fluorescence dynamics in an in vivo VSDI study of mouse primary visual cortex (V1). In particular, they find that in a localized region at the site of an applied 50 µA current impulse, fluorescence rapidly increases within ~10 ms, before saturating and then expanding laterally. They propose that the initial peak primarily reflects mono-synaptic excitatory postsynaptic potentials (EPSPs), which are followed by the propagation of disynaptic activity at greater latencies. This explanation is consistent with our finding that the first response occurs locally and quickly plateaus, likely as a consequence of feedforward PSPs evoked by direct TC innervation at the center of the microcircuit. Subsequent activity spread would occur only following a monosynaptic delay, as the targets of TC projections propagate the signal to their postsynaptic partners. Also, we note that in comparison to average signal transmission speeds reported in the literature (Supplementary Table 1), the wavefront phase velocities calculated here are relatively low. We speculate that this may be due to slicing of mid-range intracortical axons during the morphology reconstruction process. It is well known that extended axonal arbors are at risk of slicing during histological processing, and efforts were made to repair severed arbors using statistical methods[24]. However, it is unlikely that such repairs would fully correct for slicing artifacts, leaving open the possibility that significant numbers of mid-range connections are missing. If true, it would tend to decrease wavefront propagation speeds, as signal transmission would be forced to proceed strictly through short-range connections.

**Lack of correlated activity between VSDI signals and layer 2/3 PCs**. A point of disagreement between our results and those described in literature is the degree to which VSDI recordings are correlated with simultaneous whole-cell (WC) recordings in L2/3 (see Supplementary Fig. 1). Several in vivo studies have reported a high correlation between VSD fluorescence and the $V_m$ of single neurons in L2/3 rodent barrel cortex[14,16,19,34]. However, due to the technical challenges associated with simultaneously performing VSDI and WC recordings in live animals, these studies used anesthesia[14,16,19] or in vitro slice preparations[34] to establish the correspondence between $V_m$ and VSDI traces. It has been shown that anesthetic agents increase cortical synchrony and pairwise neural correlations[44–47]. Of particular relevance, Greenberg et al.[45] found that correlated AP firing in pairs and populations of L2/3 neurons in rat visual cortex increased significantly during anesthesia as compared to the awake state. Therefore, the disparity between the strength of VSDI-$V_m$ correlations observed in vivo and those extracted from our simulations may be at least partly explained by differences in cortical state. Since VSDI signals reflect an average over $V_m$ deflections in a large number of neuronal processes mostly situated in L2/3, anesthesia-induced synchrony among L2/3 neurons would tend to increase the correlation between any given L2/3 neuron and the population mean. Our model does not consider the effects of anesthesia, nor do we observe the emergence of oscillatory cortical states. Thus, both pairwise and population neural correlations remain relatively weak during spontaneous activity, resulting in a lower correspondence between VSD fluorescence and individual $V_m$ measurements.

**Changes in spiking activity precede deflections in mean membrane potential**. We showed that spikes precede $V_m$ fluctuations during both spontaneous and evoked activity (see Fig. 2c, d; Fig. 4a; Fig. 5b), confirming several studies including one in rat barrel cortex[19], and two others in ferret visual cortex[48,49]. A reasonable expectation may be that, on the contrary, increases in VSD fluorescence should precede increased spike firing, since membrane depolarization would tend to bring neurons closer to threshold making APs more likely. However, as suggested by Eriksson et al.[48], since each cell contacts many postsynaptic partners (452 ± 272 in our microcircuit), any given AP will elicit PSPs in the dendrites of hundreds to thousands of other cells. Although spike initiation requires membrane depolarization, only a fraction of the population is active at once (~26% at evoked response peak, and ~0.4% during baseline; 2 ms bins). Therefore, $V_m$ changes associated with spike firing are outweighed by downstream PSPs, with a monosynaptic delay. We found a 6.9 ms delay between peak spiking and subthreshold response during stimulation (Fig. 2c, d), and a 22.7 ms delay during spontaneous activity (Fig. 5b). Monosynaptic signal transmission reportedly requires between 6 and 14 ms in cortex[50], suggesting that deflections in mean $V_m$ primarily reflect monosynaptic activity in the first case (evoked), and disynaptic inhibition in the second (spontaneous).

**Dissecting the neuropil and the role of axonal contributions**. Though each reconstructed morphology in the NMC contains the full complement of axonal, dendritic and somatic compartments, non-axon initial segment (AIS) axonal compartments are excluded at simulation runtime to conserve resources. This is possible since axo-axonal connections are thought to be mediated by: (1) Chandelier cells which form synapses exclusively on the AIS[51,52], and (2) somatostatin-expressing cells which innervate somata, dendrites, spines, and the AIS[53], implying that non-AIS intracortical axonal arbors merely propagate signal from one point in space to another with relatively minimal disturbance. Thus, AP waveforms originating in the AIS may be broadcast to axon terminals following a suitable delay calculated from axonal path length and known conduction velocities. However, the exclusion of axonal compartments could curtail the accuracy of our VSDI model, since they are a potentially relevant source of signal. To address such concerns, we characterized the composition of the neural tissue within each voxel (1000 μm³ cube of tissue) in terms of the surface area contributed by axonal, somatic, and dendritic compartments (Supplementary Fig. 5). We found that axonal compartments constitute a non-negligible fraction of the neuropil, representing ~35% of the total membrane surface area (Supplementary Fig. 5b,d), with a peak at the boundary between layers 4 and 5 (Supplementary Fig. 5a). However, this concern must be viewed in light of two important countervailing considerations. First, electron microscopy serial reconstructions have revealed substantial, although patchy, myelination along the axonal arbors of both GABAergic and pyramidal neurons in the neocortex (Micheva et al.[54]). In particular, Tomassy et al.[55] find that up to 60% of the reconstructed axons of L5 and L6 PCs may be myelinated. This would tend to attenuate axon-related fluorescence since VSDI does not measure activity in myelinated axonal segments[1]. Second, in contrast to dendritic processes in which PSPs constantly occur as a result of synaptic bombardment, axons beyond the AIS do not receive synaptic input. Thus, each axonal compartment sees only a brief deflection in $V_m$ as an AP waveform travels along the cable. Since it is well established empirically that APs do not contribute to VSDI signals[15,16,19,34], and since neocortical axons carry mostly or only AP-related information, it follows that axons, in spite of their nontrivial total surface area, are likely of little consequence in shaping VSDI measurements. This claim awaits definitive confirmation in future modeling studies that explicitly represent axonal compartments in the calculation of in silico VSDI signals.

**Limitations and outlook**. As regards the validity of the in silico model of VSDI presented here, we considered the following three aspects: (1) accuracy of the biophysical model of VSDI, i.e., the calculations linking cellular activity to measured fluorescence, (2) biological plausibility of the composition and architecture of the anatomical tissue model, (3) whether the simulations are *functionally* representative of biological neocortex.

On the first account, we assert that the excellent linearity and fast kinetics of VSDs[22] greatly simplify their analytical relationship to $V_m$. Regarding the second concern, a caveat is the absence of several important structural details from the version of the NMC used here, including glial cells, vasculature, and long-range intracortical axons[24]. However, in the case of glia, the slow timescale of response (3–4 ms[56]) and small amplitude of $V_m$ deflections (1–7 mV[57]) make it unlikely that they contribute meaningfully to VSDI measurements. As regards vasculature, since our in silico VSDI pipeline already accounts for the bulk optical properties of cortical tissue (see "Methods"), their effects have, in principle, been accounted for. With respect to the third concern above, several details that are likely to influence network dynamics, including: gap junctions, multivesicular release, synaptic plasticity, and especially neuromodulatory dynamics, the effects of which are known to be implicated in the transitions between cortical states[58], are not present in this version of the NMC.

In addition, we note several limitations concerning the comparison of our in silico stimulation protocol to whisker deflection experiments in rodents. First, as a model of hindlimb somatosensory cortex, the NMC lacks the unique cytoarchitecture and anatomical organization that characterize barrel cortex[59,60]. Furthermore, the NMC excludes the trigeminal and thalamic nuclei, and therefore does not model sensory processing delays (and VSDI response latencies) or the modulation of cortical dynamics by TC feedback. Previous experiments have shown that cortical activation patterns depend on stimulus strength, with a tendency for excitation evoked by weak stimuli to remain confined to a single barrel[19,29,34,42]. In our model, TC projection fibers innervating the geometrical center of the microcircuit fire a simultaneous AP, a scheme that neither captures the full complexity of afferent TC signaling, nor permits a nuanced modification of stimulus strength. Finally, experiments have implicated reciprocal TC pathways[61–63] and intracortical interactions[64] in the emergence of slow wave activity. It is known that cortical oscillations interact with sensory responses to produce differentiated VSDI signals[14]. Thus, an in silico account of the effects of brain state on VSDI measurements awaits future iterations of the NMC that include TC feedback and corticocortical interactions.

Limitations aside, our digital framework permits extension from models of exogenous dye-based imaging to other modes of voltage imaging in the brain. In particular, recent advances in the development of genetically encoded voltage indicators (GEVIs) have prompted a resurgence of interest in voltage imaging. For example, Abdelfattah et al.[65] demonstrated the use of bright, photostable indicators for in vivo imaging of sub- and suprathreshold membrane voltage in populations of neurons in larval zebrafish. Our research adds important nuance to the understanding and interpretation of VSDI signals at a moment when GEVIs are poised to reinvigorate the field of voltage

imaging, and furthermore provides a foundation for future modeling studies of GEVIs and other techniques.

This study demonstrates the utility of bottom-up biophysical modeling as a complement to experimental approaches for understanding the relationships between spatial and temporal scales of cortical signaling. Using our model, we clarify which aspects of neural anatomy and physiology shape VSDI signals. Additionally, we estimate a bound on the fundamental capacity of VSDI pixels to resolve cortical activity emanating from spatially separated, point-like TC inputs. These insights were gleaned from in silico data beyond the reach of current experimental techniques.

## Methods

**Microcircuit.** Our in silico VSDI model was implemented in a digital microcircuit consisting of a connected network of 31,346 neurons, ~8 million connections, and ~37 million synapses. The network was arranged in a columnar volume 462 × 400 μm wide and 2082 μm deep. A spatially extended version was constructed by interconnecting seven such units in a hexagonal tiling (the "mosaic"). In this configuration, depth axes were mutually parallel, and columnar surfaces were coplanar (Fig. 1h). The cell morphologies populating the circuit were obtained from 3D reconstructions of biocytin-stained neurons from juvenile rat hindlimb somatosensory cortex, while the placement, connectivity, and electrophysiological properties of each cell was determined algorithmically and constrained by sparse data derived from experiments and literature[24]. TC innervation was reconstructed using VPM axonal bouton density profiles in rat barrel cortex, and synapses were probabilistically assigned to incoming fibers using a Gaussian distribution centered around each fiber[24]. See Markram et al.[24] for additional details concerning microcircuit construction and composition. Models and circuit data are freely available for download at: https://bbp.epfl.ch/nmc-portal/downloads (see "Data Availability" statement).

**Supercomputing.** A 2-rack Intel supercomputer using dual socket, 2.3 GHz, 18 core Xeon SkyLake 6140 CPUs, with a total of 120 nodes, 348 GB of memory, and 46 TB of DRAM was used to run the simulations and carry out analysis.

**Simulation.** Simulations were conducted using proprietary software based on the NEURON simulation environment[66]. Data were output in the form of binary files containing spike times and compartment $V_m$ sampled every 0.1 ms for each neuron in the network. Extracellular calcium and potassium concentrations were modeled by considering their phenomenological effects on neurotransmitter release probability and somatic depolarization, respectively. These values were adjusted to most closely mimic an in vivo-like network state[24], corresponding (empirically) to extracellular calcium and potassium concentrations of 1.25 mM, and 5.0 mM (~100% somatic firing threshold), respectively.

Trials simulating evoked responses modeled TC stimulation with a single pulse of activity in 60 contiguous thalamic fibers projecting to the geometric center of the microcircuit. For experiments requiring a larger cortical surface area, the same approach was applied to the spatially extended hexagonal microcircuit tiling (see Microcircuit above). Activity was simulated over ten trials (i.e., random seeds) for a duration of 5 s each, with the first 2 s of data in each trial discarded to avoid any boundary condition-dependent artefacts. The stimulus was delivered at 2500 ms, meaning that each trial consisted of an initial period of 500 ms of spontaneous activity, followed by 2500 ms of poststimulus activity.

**Signal calculation.** We assumed that the VSDI signal emanating from a small patch of cellular membrane was linearly related to the product of the membrane surface area and $V_m$[1,3,14,16,19,22,29,34]. Our neuronal morphologies are composed of small segments ("compartments") of equipotential cable whose surface area and transmembrane voltage were multiplied to obtain the raw VSDI signal. This raw signal was scaled for each compartment as a function of depth to account for the physics of dye diffusion and the scattering and absorption of illumination light (Fig. 1d). The degree of signal attenuation due to uneven staining through the cortical depth was interpolated from data measured in four mouse brains treated with RH1691 voltage-sensitive dye, flash-frozen, and sliced into 20 μm thick cryosections[16]. To reduce data storage requirements, we divided the microcircuit into voxels, within which an aggregate signal was computed by summing the contributions of all compartments in that voxel:

$$v_{ijk}(t) = \sum_{r \in ijk} \alpha_r \Gamma(y_r)(\Delta V_m^r(t) + G_0) \tag{2.1}$$

where $v_{ijk}$ denotes the value in the $ijk$th voxel, $\alpha_r$ is the surface area of the $r$th compartment, $\Gamma(y_r)$ is an attenuation prefactor accounting for dye penetration and scattering/absorption of illumination light at depth $y$ for compartment $r$, $\Delta V_m^r$ is the change in membrane potential for the $r$th compartment, and $G_0$ is a constant reflecting the combined contributions of background noise and autofluorescence (assumed isotropic). The value of $G_0$ was fixed by requiring that a 10 mV change in

$V_m$ correspond to a ~0.5% change in fluorescence over baseline ($\Delta F/F_0$), as reported by Ferezou et al.[16], assuming an average resting potential of −65 mV. We developed a dedicated software package (EMSim) for efficient calculation of the data volumes defined by Eq. (2.1), which is open source and freely available for download (see "Code Availability" statement)[67].

To model the effects of scattering and absorption in the tissue, we used a Monte Carlo simulation-based approach (see Point Spread Function) to compute an effective PSF for increasing depths along an axis perpendicular to the cortical surface. We used the PSF at each depth to determine the standard deviation of a Gaussian kernel, which we convolved with the horizontal data slice at that depth:

$$H_j = \begin{pmatrix} v_{0j0} \cdots v_{0jl} \vdots \ddots \vdots v_{nj0} \cdots v_{njl} \end{pmatrix} \tag{2.2}$$

$$\hat{H}_j(t) = H_j(t) * g(\sigma_j) \tag{2.3}$$

$$\hat{V} = \bigcup_j \hat{H}_j \tag{2.4}$$

$H_j$ (Eq. (2.2)) is a horizontal data slice at depth $j$, where $i \in \{0, \dots, n\}$ and $k \in \{0, \dots, l\}$. In Eq. (2.3), $\hat{H}_j$ is the filtered data slice at depth $j$, $H_j$ is the original data slice, and $g$ is a Gaussian kernel, with depth-dependent standard deviation $\sigma_j$. The union of all filtered slices yields the filtered data volume $\hat{V}$ (Eq. (2.4)). Post-convolution, each vertical ($j$-axis) column of voxels was accumulated into a single value, resulting in a two-dimensional matrix of pixels, which we stored as an image (Eq. (2.5)). VSDI signals were computed as a fractional change in fluorescence over resting intensity[3,4,17,68,69]. This gives raw and normalized signal intensities for each pixel in an image matrix:

$$F_{ik}(t) = \sum_j \hat{v}_{ijk}(t) \tag{2.5}$$

$$VSD_{ik}(t) = \frac{\Delta F_{ik}(t)}{F_{ik}^0} - 1 \tag{2.6}$$

where $F_{ik}^0$ is a baseline fluorescence image obtained by averaging the first 100 frames (50 ms of data sampled at 2000 Hz). We used Eq. (2.6) to calculate voltage-sensitive dye signals in this work. The software for the pipeline (Eqs. (2.2) through (2.6)) described above is open source and freely available for download (see "Code Availability" statement)[70].

**Point spread function.** We calculated an empirical, depth-dependent PSF to account for blurring in the final image due to both scattering of emitted fluorescence photons in cortical gray matter, and optical distortions caused by out-of-plane signal. Our method for calculating the PSF consisted of two steps: first, we used a Monte Carlo (MC) simulation-based approach to model the scattering and absorption of photons emitted from a point source within the tissue volume, varying the depth of the source in 50 micron increments; second, we used ray transfer matrix analysis to trace the trajectories of these photons through a tandem-lens optical system onto a sensor at the image plane.

MC simulations were carried out using a proprietary library built on an open-source framework for physical rendering using backward MC ray tracing, the Physically-based Rendering Toolkit (PBRT)[71]. We extended the PBRT framework to simulate photon interactions with highly turbid media using forward MC simulations based on an algorithm proposed by Abdellah et al.[72]. This software is freely available at: https://github.com/BlueBrain/pbrt-v2 (see "Code Availability" statement). To determine the PSF, we moved an isotropically radiating point source of $10^8$ photons throughout a semi-infinite (lateral extent) volume of tissue, beginning at the bottom of the microcircuit in increasing increments of 50 μm, allowing each photon to scatter until it either was absorbed or exited the cortical surface. Coefficients of reduced scattering and absorption at ~665 nm were taken to be 4 mm$^{-1}$ and 0.4 mm$^{-1}$, respectively, interpolated from optical measurements made in rat gray matter for wavelengths of light spanning 450 to 700 nm[73]. We chose the wavelength to correspond to peak emissions in the RH-1691/1692 family of blue voltage-sensitive dyes[4,14,16,34]. Refraction at the tissue-air interface was calculated using the vector formulation of Snell's law. Using ray transfer matrix analysis, photons emanating from the tissue surface were propagated through an optical system modeled after a tandem-lens epifluorescence macroscope setup first proposed by Ratzlaff and Grinvald[74], and subsequently used in several VSDI studies[14,16,19]. The system consists of two compound lenses (modeled using the thin lens approximation) set to infinite focus and placed face-to-face[74]. Optical parameters (focal length, f-number, and working distance) were taken from Petersen et al.[75], resulting in a focal plane ~300 μm below the pia. The point source produced a sunburst image pattern on the detector array for each depth, to which a two-dimensional Gaussian surface was fit using a nonlinear optimizer (Python). From these surfaces, we extracted the average spatial standard deviation, and fit the resulting array of values to a decaying exponential function to determine a depth-dependent PSF for the entire tissue-lens system. The standard deviations extracted from our PSF were used to calculate spatial kernel widths for convolution of the data with a Gaussian filter (see Eq. (2.3)).

**Two-point discrimination**. We simulated multiple trials of a localized, point-like stimulus in the NMC (a single AP in five adjacent TC VPM projection fibers), for increasing distances from the geometrical center of the circuit, repeated fives times at intervals of 50 ms. Stimuli were delivered to locations at −50, −25, −12.5, 0, 12.5, 25, and 50 μm along a line bisecting the surface of the NMC. Repeated trials ($n = 25$) were simulated for each stimulus location.

*Spike train-based classification*. We carried out a receiver operating characteristic (ROC) analysis to quantify the ability of the spike trains of individual neurons to discriminate stimulus locations. The performance of each neuron was compared to all others within a subpopulation defined by morphology type. Our classification procedure considered whether spike count in the 250 ms post-stimulus window crossed a threshold that was selected to optimize classification performance of the highest performing neuron in each subpopulation considered. We formed all 50 choose 2 combinations (paired data, $n = 25$ trials per condition) of spike train data for each neuron and evaluated whether the post-stimulus spike count was above or below the empirical threshold for that subpopulation. Since each of the 50 choose 2 pairings comprised either two trials from different stimulus locations, or two from the same location, threshold-based classification corresponded to either a true positive, or false positive result, respectively. Thus, we were able to assign to each neuron a true positive (TPR) and false positive (FPR) classification rate. Finally, integrating the resulting ROC curves yielded AUROC scores for each neuron.

*Time-resolved pixel-based classification*. For each combination of two stimuli, for each VSDI pixel, and for each timestep, we used a Wilcoxon rank-sum test to compute the p-value of the null hypothesis that the data (pooled over $n = 25$ trials for each location, respectively) associated with the two stimulus positions came from the same distribution. We defined the "discrimination power" of a given pixel for a particular stimulus pairing as the negative common logarithm of the p-value of the null hypothesis stated above.

*Temporally integrated pixel-based classification*. Similar to our study of spike train-based stimulus classification, we used an ROC analysis to quantify the classification performance of the time series data (considered as a whole) of individual VSDI pixels. To this end, we used a Wilcoxon rank-sum test for each pair of stimulus locations to compute the p-value of the null hypothesis that the time series data associated with each location came from the same distribution. Stimulus locations were considered distinct if the p-value produced by comparing the two pools of VSD fluorescence data for any given combination of trials was above a predetermined significance threshold. For the ROC curve, we observed how varying the significance threshold altered the proportion of total true positive rejections of the null hypothesis. As in the spike train-based classification, we formed all 50 choose 2 combinations of time series pixel data, and calculated a TPR and FPR for each significance threshold to produce a ROC curve. Integrating this curve generated an AUROC score for each pixel and location pairing.

*Spatially and temporally integrated classification*. To assess the classification accuracy of VSDI movies considered as a whole, we pooled the data for all $n = 25$ trials of a given stimulus location and formed an aggregate dataset of 50 trials for each possible combination of stimulus locations (7 choose 2). We flattened each time series of VSDI images into a one-dimensional vector and stacked the results into a single $2n \times Tl^2$ matrix:

$$X = \begin{pmatrix} p_{n=0,0}^{0,0,0} \cdots p_{n=0,0}^{T,l,l} \vdots \vdots p_{n=24,0}^{0,0,0} \cdots p_{n=24,0}^{T,l,l} \\ p_{n=0,1}^{0,0,0} \cdots p_{n=0,1}^{T,l,l} \vdots \vdots p_{n=24,1}^{0,0,0} \cdots p_{n=24,1}^{T,l,l} \end{pmatrix} \qquad (3.1)$$

where $n$ is the number of trials per condition, $T$ is the number of post-stimulus time steps, and $l$ is the side length of the optical sensor. Each of the $n = 50$ rows was assigned a label corresponding to the location of the stimulus:

$$y = \begin{pmatrix} 0 \\ \vdots \\ 0 \\ 1 \\ \vdots \\ 1 \end{pmatrix} \qquad (3.2)$$

We used an open-source of implementation the XGBoost algorithm for Python (available at: https://pypi.org/project/xgboost/) to train a binary classifier on the data $X$ and $y$ described above. Train and test sets were randomly partitioned using an 80–20 split (fivefold cross-validation), and accuracy was computed as the fraction of correct predictions:

$$\text{accuracy} = \frac{1}{n} \sum_{i=0}^{n-1} 1(\hat{y}_i = y_i) \qquad (3.3)$$

**Post-processing and analysis**. All code for analysis was written in the Python programming language.

**Reporting summary**. Further information on research design is available in the Nature Research Reporting Summary linked to this article.

## Data availability
The models and circuit data used for this study are freely available for download at: https://bbp.epfl.ch/nmc-portal/downloads. The raw, post-processed data required for reproduction of all manuscript figures (main and supplementary) are freely available for download at: https://doi.org/10.5281/zenodo.4733519. All figure data (main and supplementary) are freely available in tabular format at: https://doi.org/10.5281/zenodo.4733519. The raw, pre-processed data, i.e., raw compartment voltages and VSDI volumetric datasets, are available upon request. Source data are provided with this paper.

## Code availability
EMSim, a software package for the efficient calculation of data volumes (see Eq. (2.1)) resulting from electromagnetic field-dependent biophysical signals is freely available at: https://doi.org/10.5281/zenodo.472557[67]. All code pertaining to the in silico VSDI pipeline (see Eqs. (2.2) through (2.6)) is open source and available at: https://doi.org/10.5281/zenodo.472548[70]. Our extension of the Physically-based Rendering Toolkit (PBRT)[71] for computing Monte Carlo simulations of photon interactions in highly turbid media is freely available at: https://github.com/BlueBrain/pbrt-v2. All analysis code (i.e., code used to generate figures from VSD imaging datasets, together with the datasets themselves) is freely available for download at: https://doi.org/10.5281/zenodo.4733519[76].

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

## Acknowledgements

We thank M. Nolte, G. Chindemi, S. Ramaswamy, C. Colangelo, and other members of the Blue Brain Project for useful feedback and discussions, and C. Favreau for visualization support. The authors are also grateful to J. King, M. Gevaert, and W. Van Geit for technical assistance. This study was supported by funding to the Blue Brain Project, a research center of the École Polytechnique Fédérale de Lausanne, from the Swiss government's ETH Board of the Swiss Federal Institutes of Technology.

## Author contributions

T.H.N., E.B.M., and H.M. conceptualized the study. T.H.N. designed and carried out the simulations, analyzed the data (excepting two-point discrimination), prepared the figures, and wrote the manuscript. M.W.R. designed and carried out the two-point discrimination procedure and analyzed the data. M.A. performed the Monte Carlo simulations of photons in brain tissue. G.C. developed the software for voxel-based calculation of the VSDI signal from membrane voltage data in consultation with T.H.N.

## Competing interests

The authors declare no competing interests.
