## [Peer Review File · Nature Communications]

Reviewer #1 (Remarks to the Author):

The authors describe an extensive biophysical model that depicts voltage sensitive dye imaging in vivo. Here, they have taken data from the neocortical microcircuit which involves synaptic conductances and thalamic input to realistically model cortical activation and how this relates to voltage sensitive dye signals. The work is useful since voltage sensitive dye imaging can be difficult to interpret given the dye's rather promiscuous diffusion and inability to be targeted to specific cell types. Here, by building this realistic cortical model, the authors can better examine the role of specific cell populations in cortical excitability and aggregate voltage signals. The work is surprising for showing that cortical excitation due to action potentials occurs appreciably before voltage sensitive dyes signals are detected. Furthermore, they show an anti-correlation between period of cortical spiking and voltage sensitive dye depolarizing transients.

Concerns.

While the work is well-done from a biophysical modeling standpoint and contain sophisticated data presentation and statistics, it suffers from having not adequately described well known results from the literature where ephys and voltage sensitive dye measures were made, in this manner the model is clearly not sufficient to describe the data.

These observations include the expected positive correlation between population spiking and voltage sensitive dye signals. While the authors do offer up some explanations concerning recurrent inhibition, my feeling is that there is probably something missing within this model which leads to these anti-correlations.

My suggestion would be for the authors to rely on more conventional aspects of their findings, including the lateral spread of voltage signals through cortex. Other model data which is relatively strong include the modeling of optical properties and dye penetration. If the authors can remove the contentious conclusions from the paper (spiking and VSD latency and anti-correlation), it would still be a significant conclusion and would be less likely to mislead the field.

The authors' should comment on extension of the model to genetically encoded sensors.

The authors comment about using realistic concentrations of calcium in the model, yet use 5 mM K⁺ to generate expected excitability. This needs to be justified as this concentration is higher than physiological levels.

Reviewer #2 (Remarks to the Author):

Review of "Voltage-sensitive dye imaging reveals inhibitory modulation of ongoing cortical activity" by Taylor H. Newton, Marwan Abdellah, Grigori Chevtchenko, Eilif B. Muller, and Henry Markram submitted to Nature Communications

Newton and colleagues present a computational model of the voltage sensitive dye imaging (VSDI) signal generated from a previously-published detailed model of a region ($\sim 1\text{mm}^2$) of rodent somatosensory cortex (Markram et al., 2015), validate the model with respect to experimental results, and make use of the model to explore circuit properties during spontaneous and evoked activity. The main effects considered in the VSDI model are dye penetration into the tissue (interpolated from experimental data) and light scattering/absorption (using Monte Carlo

simulations of 'photon interactions with highly turbid media'). Simulation of a pulse of thalamic activity produces an expanding wave of activity, with the time to peak and the half-width giving good agreement with experimental findings, but the propagation speed of the wave being lower than most experimentally observed values. Following this validation, the authors used the model first to confirm the experimental finding that the majority of the VSDI signal is generated within 500 μm of the pial surface, and then moreover to show that excitatory neurons in Layer 5 contribute the majority of the signal during evoked activity, with a large contribution also from excitatory neurons in Layer 2/3. The smaller contribution from the more superficial neurons is explained by a tendency to post-stimulus hyperpolarization of the L2/3 neurons.

Further analyses and simulations were used to: (i) estimate the relative contributions of action potentials vs sub-threshold depolarization to the VSDI signal;(ii) investigate the somewhat counterintuitive finding that the peak of action potential firing summed over the entire circuit occurs earlier than the peak of the VSDI signal (reflecting membrane potential), resulting in a finding that population firing rates and mean membrane potential are anticorrelated during spontaneous activity, this being due to a preponderance of inhibitory feedback in driving the VSDI signal. Finally, the authors modified the model to partially address one of its limitations, replacing somatic noise injection with additional synaptic inputs in a small subset of neurons, with the conclusion that this limitation (due to resource constraints) is an important one that needs to be addressed in future work. The manuscript thoroughly discusses the validity and implications of the results, and the limitations of the model. All statistical analyses appears to be valid, although in any case the effect sizes for the main results are all large, and so tests of significance are not really essential in supporting the conclusions.

In general, I found this to be a convincing and well-performed study that will provide inspiration and insights for both experimental and computational neuroscientists. It is a good demonstration of how the enormous effort that goes into building a highly detailed model of neural circuits (Markram et al., 2015) can be exploited to elucidate mechanisms, shed light on open questions, and inspire further studies. In particular I found the replacement of recurrent connections with activity recorded from previous simulations with different random seeds to be rather elegant.

However, I have a number of concerns that should be addressed before publication.

1. I am confused that the VSDI signal shown in Fig 2b is monophasic and has an amplitude of $\sim 2.5\%$, but that shown in Figures 3f and 4a is biphasic with amplitude $\sim 1\%$, although as far as I can see from the text, the same stimulus and spatio-temporal filtering are used in the two cases. Presumably there is some difference which explains this, but I could not find it explained anywhere.

2. In the presentation of the simulations of stimulus-evoked activity, it would be helpful if the breakdown of firing rate responses for individual layers and for excitatory vs inhibitory neurons were shown, perhaps in an additional supplementary figure. For example, the finding that peak AP firing precedes peak fluorescence is probably only counter-intuitive when lumping the firing of all neuronal sub-populations together.

3a. I do not think this work can be reproduced given the information and materials provided. I understand that the connectivity data and neuron models underlying the NMC model can be downloaded from the NMC Portal of the Blue Brain Project. However, none of the code or data for modelling the VSDI signal is given, and there is no statement on model and data sharing. I strongly encourage the authors to deposit all code/data in a public repository such as ModelDB, OpenSourceBrain or the EBRAINS model catalog, following the guidelines at

<http://opensourceforneuroscience.org>. If the proprietary software used for the Monte Carlo simulations cannot be released under an open source licence, a statement about how it may be obtained (e.g. through purchase, or on request) should be provided.

3b. On the principle that a modelling study should be reproducible from the paper even in the absence of source code, the following information is missing from the Methods: voxel size; an expression for or tabulation of the attenuation function $\Gamma(y)$; the value of G_0 ; an expression for or tabulation of the standard deviation of the PSF kernel $\sigma_j(y)$.

Minor issues:

- a. The abbreviation "TC" is never defined (presumably "thalamocortical" or "thalamocortical fiber")
- b. Figure 3k. The meaning of "***" is not explained anywhere.
- c. Figure 3 caption: "same as c" should be "same as d"; "same as d" should be "same as e"
- d. Figure 4, caption: "top" and "bottom" are inverted with respect to the figure.
- e. Figure 4. the pale lines are not explained (presumably individual trials)
- f. Figure 4. inconsistent use of "SNR / signal-to-noise" and "SSR" in the figure and caption. For example, the legend is "SSR", but the y-axis label is "SNR". It should presumably be "signal-to-spike" ratio in all cases.
- g. "In a hypothetical a scenario" - superfluous 'a'
- h. In the final paragraph before the concluding remarks, the manuscript refers to "forthcoming" models, then references papers published last year
- i. "stimulation with of a single pulse" - superfluous 'of'
- j. "reflecting the combined reflecting the combined" - duplication
- k. Figure 3b. The y-axis seems to be inverted, if we assume that depth 0 should occur at the surface.
- l. Supplementary Figure 2. What are "Bio-1..M"?

Signed,

Andrew P. Davison
Paris-Saclay Institute of Neuroscience
CNRS / Université Paris-Saclay

February 11th 2020

Reviewer #3 (Remarks to the Author):

This manuscript exploits a previously developed, large-scale, biophysically detailed network model of a cortical column to study the relationship between the signal recorded using voltage-sensitive dyes and the underlying neural activity. To this end, the authors have extended the existing model to include the response dynamics of voltage-sensitive dyes (VSD). They then examined which aspects of neural activity determine the VSD signal, both in spontaneous and evoked dynamics.

To me, this study beautifully demonstrates how a large scale network model can be used to address biological questions that are otherwise inaccessible. Overall the manuscript is well written, and the figures well presented.

My only concern is the section "VSDI signals anticorrelate with population firing rates", which I find quite difficult to follow. Specifically:

- in the end of the first paragraph, the authors introduce a mean-field approach. However, as far as I can see from the text and the Methods, the mean-field theory models only the relationship between conductance and membrane potential, while the goal here is to understand the relationship between membrane potential and firing rate. That piece seems to be missing, and without it the aim of the mean-field analysis is unclear.

- I don't understand Fig 5e. What is shown there, and what is the point?

- the sequence of arguments in the second paragraph of that section is difficult to follow. The authors first insist on E-I balance, but it seems that what matters for the anti-correlation is that inhibition dominates. I don't quite understand how "it follows that mean V , and by extension the VSDI signal, anticorrelate with total firing rate". Again an argument relating the total firing rate with the membrane potential seems to be missing. The synaptic timescale presumably play a crucial role, but are not discussed at all.

- in contrast, the explanation in the Discussion is much more clear (section "Changes in spiking activity precede deflections in mean membrane potential"). In particular, the delay in the anti-correlations is explained there, but not in the main text. I would therefore suggest using arguments from that part of the Discussion in the Results section "VSDI signals anticorrelate with population firing rates".

- It would be useful to show the time dynamics of excitatory and inhibitory conductances in Fig.5a.

- Fig.5f and Fig.5j seem redundant.

**Reviewer #1 (Remarks to the Author):**

The authors describe an extensive biophysical model that depicts voltage sensitive dye
imaging in vivo. Here, they have taken data from the neocortical microcircuit which
involves synaptic conductances and thalamic input to realistically model cortical
activation and how this relates to voltage sensitive dye signals. The work is useful since
voltage sensitive dye imaging can be difficult to interpret given the dye's rather
promiscuous diffusion and inability to be targeted to specific cell types. Here, by building
this realistic cortical model, the authors can better examine the role of specific cell
populations in cortical excitability and aggregate voltage signals. The work is surprising
for showing that cortical excitation due to action potentials occurs appreciably before
voltage sensitive dyes signals are detected. Furthermore, they show an anti-correlation
between period of cortical spiking and voltage sensitive dye depolarizing transients.

Concerns.

While the work is well-done from a biophysical modeling standpoint and contain
sophisticated data presentation and statistics, it suffers from having not adequately
described well known results from the literature where ephys and voltage sensitive dye
measures were made, in this manner the model is clearly not sufficient to describe the
data.

These observations include the expected positive correlation between population spiking
and voltage sensitive dye signals. While the authors do offer up some explanations
concerning recurrent inhibition, my feeling is that there is probably something missing
within this model which leads to these anti-correlations.

My suggestion would be for the authors to rely on more conventional aspects of their
findings, including the lateral spread of voltage signals through cortex. Other model data
which is relatively strong include the modeling of optical properties and dye penetration.
If the authors can remove the contentious conclusions from the paper (spiking and VSD
latency and anti-correlation), it would still be a significant conclusion and would be less
likely to mislead the field.

Upon further reflection, we agree that our claim regarding the relationship between
membrane potential and spiking is sufficiently surprising (in light of expectations and the
weight of available evidence) as to require more proof of its existence. Therefore, as per
your suggestion, we have decided to remove the figures and text pertaining to the
stated anticorrelation, and simulation of additional extrinsic synapses. Instead, we focus
on the more conventional aspects of our findings, including the lateral propagation of
VSDI activity, and signal contributions by layer. In addition, we offer a new analysis that
examines the ability of individual VSDI pixels to discriminate between spatially separated
but otherwise identical thalamic inputs.

The authors' should comment on extension of the model to genetically encoded sensors.

Please see Discussion section for requested comments.

The authors comment about using realistic concentrations of calcium in the model, yet
use 5 mM K⁺ to generate expected excitability. This needs to be justified as this
concentration is higher than physiological levels.

Thank you for raising this issue. Our previous wording (i.e., referring to extracellular
potassium concentration as a “free parameter”) was misleading. To be more precise, we
modeled extracellular calcium and potassium phenomenologically by considering the
effects on synaptic release probability and somatic depolarization, respectively. In
Markram et al.*, we highlight the emergence of a spectrum of network states that
depends on these two parameters (synaptic release and somatic depolarization). In that
study, we found that an “*in vivo-like*” critical point exists on this spectrum, where the
network just transitions from a regime of asynchronous, sparse spiking, to one of
synchronous bursting. This critical point happens to occur in our model at a synaptic
release probability and somatic depolarization that correspond to an extracellular
calcium concentration of 1.25 mM, and an extracellular potassium concentration of 5.0
mM. Thus, the values were defined by a sensitivity analysis of the model itself, rather
than being free parameters taken from literature. We’ve altered the text in Methods to
clarify this point:

*“Extracellular calcium and potassium concentrations were modeled by considering their*
*phenomenological effects on neurotransmitter release probability and somatic*
*depolarization, respectively. These values were adjusted to most closely mimic an in vivo-*
*like network state (Markram et al., 2015), corresponding (empirically) to extracellular*
*calcium and potassium concentrations of 1.25 mM, and 5.0 mM (~100% somatic firing*
*threshold), respectively.”*

*Markram, H., Muller, E., Ramaswamy, S., Reimann, M.W., Abdellah, M., Sanchez, C.A.,
Ailamaki, A., Alonso-Nanclares, L., Antille, N., Arsever, S., et al. (2015). Reconstruction and
Simulation of Neocortical Microcircuitry. Cell 163, 456–492.

**Reviewer #2 (Remarks to the Author):**

Review of "Voltage-sensitive dye imaging reveals inhibitory modulation of ongoing
cortical activity"

by Taylor H. Newton, Marwan Abdellah, Grigori Chevtchenko, Eilif B. Muller, and Henry
Markram

submitted to Nature Communications

Newton and colleagues present a computational model of the voltage sensitive dye
imaging (VSDI) signal generated from a previously-published detailed model of a region
($\sim 1\text{mm}^2$) of rodent somatosensory cortex (Markram et al., 2015), validate the model
with respect to experimental results, and make use of the model to explore circuit
properties during spontaneous and evoked activity. The main effects considered in the
VSDI model are dye penetration into the tissue (interpolated from experimental data)
and light scattering/absorption (using Monte Carlo simulations of 'photon interactions
with highly turbid media'). Simulation of a pulse of thalamic activity produces an
expanding wave of activity, with the time to peak and the half-width giving good
agreement with experimental findings, but the propagation speed of the wave being
lower than most experimentally observed values. Following this validation, the authors
used the model first to confirm the experimental finding that the
majority of the VSDI signal is generated within 500 μm of the pial surface, and then
moreover to show that excitatory neurons in Layer 5 contribute the majority of the signal
during evoked activity, with a large contribution also from excitatory neurons in Layer
2/3. The smaller contribution from the more superficial neurons is explained by a
tendency to post-stimulus hyperpolarization of the L2/3 neurons.

Further analyses and simulations were used to: (i) estimate the relative contributions of
action potentials vs sub-threshold depolarization to the VSDI signal;(ii) investigate the
somewhat counterintuitive finding that the peak of action potential firing summed over
the entire circuit occurs earlier than the peak of the VSDI signal (reflecting membrane
potential), resulting in a finding that population firing rates and mean membrane
potential are anticorrelated during spontaneous activity, this being due to a
preponderance of inhibitory feedback in driving the VSDI signal. Finally, the authors
modified the model to partially address one of its limitations, replacing somatic noise
injection with additional synaptic inputs in a small subset of neurons, with the conclusion
that this limitation (due to resource constraints) is an important one that needs to be
addressed in future work. The manuscript thoroughly discusses the validity and
implications of the results, and the
limitations of the model. All statistical analyses appears to be valid, although in any case
the effect sizes for the main results are all large, and so tests of significance are not really
essential in supporting the conclusions.

In general, I found this to be a convincing and well-performed study that will provide
inspiration and insights for both experimental and computational neuroscientists. It is a
good demonstration of how the enormous effort that goes into building a highly
detailed model of neural circuits (Markram et al., 2015) can be exploited to elucidate
mechanisms, shed light on open questions, and inspire further studies. In particular I
found the replacement of recurrent connections with activity recorded from previous
simulations with different random seeds to be rather elegant.

However, I have a number of concerns that should be addressed before publication.

1. I am confused that the VSDI signal shown in Fig 2b is monophasic and has an
amplitude of ~2.5%, but that shown in Figures 3f and 4a is biphasic with amplitude ~1%,
although as far as I can see from the text, the same stimulus and spatio-temporal
filtering are used in the two cases. Presumably there is some difference which explains
this, but I could not find it explained anywhere.

Explanation added- please see footnote added to Results section: "Excitatory neurons in
layers 2/3 and 5 dominate VSDI measurements". Here is the text:

*Note that the analysis of data for Fig. 3 was performed in an isolated NMC column. In
contrast, data in Fig. 2 was computed in a mosaic concatenation of 7 NMCs to allow
sufficient surface area for signal spread. This accounts for slight differences in peak activity
profiles (monophasic vs. biphasic) and peak amplitudes (~2.5% vs. ~1%) visible in Figs. 2b
and 3f.*

2. In the presentation of the simulations of stimulus-evoked activity, it would be helpful if
the breakdown of firing rate reponses for individual layers and for excitatory vs inhibitory
neurons were shown, perhaps in an additional supplementary figure. For example, the
finding that peak AP firing precedes peak fluorescence is probably only counter-intuitive
when lumping the firing of all neuronal sub-populations together.

Please see the supplementary materials for the additional requested figure (firing rate by
layer and EXC/INH neurons). A detailed consideration of the breakdown of firing rates
into layers and neuron class does not account for the relative delay in peak VSD
fluorescence. We believe, as stated in the manuscript, that the effect is a consequence of
the fact that a single AP causes hundreds or thousands of PSPs. Since VSD primarily
measures voltage changes in dendrites, it follows that a peak in the firing of APs would
tend to produce a peak in the VSD signal, following a monosynaptic delay. This has
been confirmed empirically in ferret cortex (Roland et al. (2006), Eriksson et al. (2008);
see Discussion).

3a. I do not think this work can be reproduced given the information and materials
provided. I understand that the connectivity data and neuron models underlying the
NMC model can be downloaded from the NMC Portal of the Blue Brain Project.

However, none of the code or data for modelling the VSDI signal is given, and there is
no statement on model and data sharing. I strongly encourage the authors to deposit all
code/data in a public repository such as ModelDB, OpenSourceBrain or the EBRAINS

model catalog, following the guidelines at <http://opensourceforneuroscience.org>. If the
proprietary software used for the Monte Carlo simulations cannot be released under an
open source licence, a statement about how it may be obtained (e.g. through purchase,
or on request) should be provided.

We understand this concern and agree the code should be available. Therefore, we have
undertaken the effort to open-source all code we used to generate the VSDI datasets
(links also provided in Methods section):

- 1) Blue Brain Project models and circuit data:
<https://bbp.epfl.ch/nmc-portal/downloads>
- 2) EMSim software, used to compute VSD data volumes from simulation data:
<https://github.com/BlueBrain/EMSim>
- 3) Monte Carlo Simulations (based on the Physically Based Rendering Toolkit):
<https://github.com/BlueBrain/pbrt-v2>
- 4) VSDI pipeline, proprietary software described in the paper which renders data
volumes produced by EMSim into VSD movies:
<https://github.com/BlueBrain/insilico-vsdi>

Please see Methods for additional descriptions and links to the software.

3b. On the principle that a modelling study should be reproducible from the paper even
in the absence of source code, the following information is missing from the Methods:
voxel size; an expression for or tabulation of the attenuation function $\Gamma(y)$; the value of
G_0 ; an expression for or tabulation of the standard deviation of the PSF kernel $\sigma_j(y)$.
Please see the supplementary materials for a table containing the values of several key
parameters, including G_0 , u_{eff} (Beer-Lambert law effective extinction coefficient for
attenuation of light in tissue), optical sensor resolution and dimensions, mean resting
membrane potential, VSDI acquisition sampling frequency, and times used to compute a
blank (baseline) fluorescence reference frame for normalization. Regarding the
attenuation function $\Gamma(y)$, the PSF kernel $\sigma_j(y)$, and several other functions, please refer
to the VSD pipeline code and associated documentation which will be released in open-
source format along with the resubmission of this manuscript.

Minor issues:

a. The abbreviation "TC" is never defined (presumably "thalamocortical" or
"thalamocortical fiber")

Fixed. Line 93: "Our stimulation protocol consisted of a single pulse of activity in 60
contiguous thalamocortical (TC) fibers emanating from a virtual ventral..."

b. Figure 3k. The meaning of "***" is not explained anywhere.

Fixed (see caption).

c. Figure 3 caption: "same as c" should be "same as d"; "same as d" should be "same as
e"

Fixed.

224 d. Figure 4, caption: "top" and "bottom" are inverted with respect to the figure.

Fixed.

e. Figure 4. the pale lines are not explained (presumably individual trials)

Fixed.

f. Figure 4. inconsistent use of "SNR / signal-to-noise" and "SSR" in the figure and

caption. For example, the legend is "SSR", but the y-axis label is "SNR". It should

presumably be "signal-to-spike" ratio in all cases.

Fixed.

232 g. "In a hypothetical a scenario" - superfluous 'a'

Fixed (text removed).

234 h. In the final paragraph before the concluding remarks, the manuscript refers to

235 "forthcoming" models, then references papers published last year

Fixed.

i. "stimulation with of a single pulse" - superfluous 'of'

Fixed.

j. "reflecting the combined reflecting the combined" – duplication

Fixed.

k. Figure 3b. The y-axis seems to be inverted, if we assume that depth 0 should occur at

the surface.

A depth of 0 corresponds by convention in our model to the bottom of the NMC, at the

interface with white matter.

245 l. Supplementary Figure 2. What are "Bio-1..M"?

Fixed (description included in caption).

Signed,

Andrew P. Davison

Paris-Saclay Institute of Neuroscience

CNRS / Université Paris-Saclay

February 11th 2020

**Reviewer #3 (Remarks to the Author):**

This manuscript exploits a previously developed, large-scale, biophysically detailed
network model of a cortical column to study the relationship between the signal
recorded using voltage-sensitive dyes and the underlying neural activity. To this end, the
authors have extended the existing model to include the response dynamics of voltage-
sensitive dyes (VSD). They then examined which aspects of neural activity determine the
VSD signal, both in spontaneous and evoked dynamics.

To me, this study beautifully demonstrates how a large scale network model can be used
to address biological questions that are otherwise inaccessible. Overall the manuscript is
well written, and the figures well presented.

My only concern is the section "VSDI signals anticorrelate with population firing rates",
which I find quite difficult to follow. Specifically:

- in the end of the first paragraph, the authors introduce a mean-field approach.

However, as far as I can see from the text and the Methods, the mean-field theory
models only the relationship between conductance and membrane potential, while the
goal here is to understand the relationship between membrane potential and firing rate.
That piece seems to be missing, and without it the aim of the mean-field analysis is
unclear.

- I don't understand Fig 5e. What is shown there, and what is the point?

- the sequence of arguments in the second paragraph of that section is difficult to
follow. The authors first insist on E-I balance, but it seems that what matters for the anti-
correlation is that inhibition dominates. I don't quite understand how "it follows that
mean V , and by extension the VSDI signal, anticorrelate with total firing rate". Again an
argument relating the total firing rate with the membrane potential seems to be missing.
The synaptic timescale presumably play a crucial role, but are not discussed at all.

- in contrast, the explanation in the Discussion is much more clear (section "Changes in
spiking activity precede deflections in mean membrane potential"). In particular, the
delay in the anti-correlations is explained there, but not in the main text. I would
therefore suggest using arguments from that part of the Discussion in the Results
section "VSDI signals anticorrelate with population firing rates".

- It would be useful to show the time dynamics of excitatory and inhibitory conductances
in Fig.5a.

- Fig.5f and Fig.5j seem redundant.

As per the suggestion of Reviewer #1, we have decided to remove the figures and text
pertaining to the stated anticorrelation between VSDI signals and membrane potential,
and simulation of additional extrinsic synapses. Instead, we focus on the more
conventional aspects of our findings, including the lateral propagation of VSDI activity,
and signal contributions by layer. In addition, we offer a new analysis that examines the
ability of individual VSDI pixels to discriminate between spatially separated but otherwise
identical thalamic inputs.

Reviewer #1 (Remarks to the Author):

This is a well-crafted bottom up model. The authors have addressed most of my concerns. I think it is important to clearly state the value of this model. There is a conclusion in the abstract about the resolution of voltage sensitive dye imaging. It seems important but no numbers are given in the abstract. When you go to the manuscript you find that a somewhat vague number is given, in this case the authors indicate that the resolution limit is somewhat greater than 25 micrometers. What does this estimate actually mean? Can they be a bit more confident in these values and give an upper and lower range? I dont think anyone would think the resolution could be better than 25 um?

Reviewer #2 (Remarks to the Author):

The revised manuscript and the links to open-source software together satisfy the concerns I had about the first version.

Signed,

Andrew P. Davison
Paris-Saclay Institute of Neuroscience
CNRS / Université Paris-Saclay
December 4th 2020

**Reviewer #1 (Remarks to the Author):**

This is a well-crafted bottom up model. The authors have addressed most of my
concerns. I think it is important to clearly state the value of this model. There is a
conclusion in the abstract about the resolution of voltage sensitive dye imaging. It
seems important but no numbers are given in the abstract. When you go to the
manuscript you find that a somewhat vague number is given, in this case the authors
indicate that the resolution limit is somewhat greater than 25 micrometers. What
does this estimate actually mean? Can they be a bit more confident in these values
and give an upper and lower range? I dont think anyone would think the resolution
could be better than 25 um?

Thank you again for your feedback. Indeed, the resolution of VSDI is unlikely to be
better than 25 um, but we still feel there is value in providing a specific, model-based
prediction for the “best case scenario” resolution attainable using available
technology. As stated in the manuscript text:

*“We further found that discrimination performance declined for smaller separations,*
*dropping sharply at approximately 25 μm (Fig. 5e). We interpret this value as a lower*
*bound on the effective input resolution of VSD imaging, accounting for both the*
*aggregate nature of the signal and optical distortions introduced by photon scattering*
*and absorption in the tissue. In most laboratory settings, additional sources of noise*
*will also contaminate the signal, further compromising discrimination between input*
*stimulus locations.”*

In addition, we have added the following sentence to the results section, which we
hope will provide further clarification for the reader (lines 352-354):

*“Thus, future advances in imaging technology are unlikely to improve VSDI’s*
*fundamental capacity to resolve loci of cortical activity below this limit.”*

**Reviewer #2 (Remarks to the Author):**

The revised manuscript and the links to open-source software together satisfy the
concerns I had about the first version.

Signed,

Andrew P. Davison
Paris-Saclay Institute of Neuroscience
CNRS / Université Paris-Saclay
December 4th 2020

Thank you for your feedback.